# CT-Derived Body Composition Is a Predictor of Survival after Esophagectomy

**DOI:** 10.3390/jcm12062106

**Published:** 2023-03-08

**Authors:** Kartik Iyer, Cameron A. Beeche, Naciye S. Gezer, Joseph K. Leader, Shangsi Ren, Rajeev Dhupar, Jiantao Pu

**Affiliations:** 1Department of Radiology, University of Pittsburgh School of Medicine, Pittsburgh, PA 15213, USA; 2Department of Cardiothoracic Surgery, Division of Thoracic and Foregut Surgery, School of Medicine, University of Pittsburgh, Pittsburgh, PA 15213, USA; 3Surgical Services Division, Thoracic Surgery, VA Pittsburgh Healthcare System, Pittsburgh, PA 15213, USA; 4Department of Bioengineering, University of Pittsburgh, Pittsburgh, PA 15213, USA

**Keywords:** esophagectomy, esophageal cancer, body composition, radiomics, survival

## Abstract

Background: Body composition can be accurately quantified based on computed tomography (CT) and typically reflects an individual’s overall health status. However, there is a dearth of research examining the relationship between body composition and survival following esophagectomy. Methods: We created a cohort consisting of 183 patients who underwent esophagectomy for esophageal cancer without neoadjuvant therapy. The cohort included preoperative PET-CT scans, along with pathologic and clinical data, which were collected prospectively. Radiomic, tumor, PET, and body composition features were automatically extracted from the images. Cox regression models were utilized to identify variables associated with survival. Logistic regression and machine learning models were developed to predict one-, three-, and five-year survival rates. Model performance was evaluated based on the area under the receiver operating characteristics curve (ROC/AUC). To test for the statistical significance of the impact of body composition on survival, body composition features were excluded for the best-performing models, and the DeLong test was used. Results: The one-year survival model contained 10 variables, including three body composition variables (bone mass, bone density, and visceral adipose tissue (VAT) density), and demonstrated an AUC of 0.817 (95% CI: 0.738–0.897). The three-year survival model incorporated 14 variables, including three body composition variables (intermuscular adipose tissue (IMAT) volume, IMAT mass, and bone mass), with an AUC of 0.693 (95% CI: 0.594–0.792). For the five-year survival model, 10 variables were included, of which two were body composition variables (intramuscular adipose tissue (IMAT) volume and visceral adipose tissue (VAT) mass), with an AUC of 0.861 (95% CI: 0.783–0.938). The one- and five-year survival models exhibited significantly inferior performance when body composition features were not incorporated. Conclusions: Body composition features derived from preoperative CT scans should be considered when predicting survival following esophagectomy.

## 1. Introduction

Approximately 90% of cancer-related mortality following esophagectomy occurs due to distant disease. The current approach to predict survival primarily relies on pathologic or clinical staging and response to neoadjuvant treatments [1]. Identifying factors associated with postoperative survival may provide valuable guidance to clinicians and patients regarding treatment and prognosis [2]. Radiographically derived variables and machine learning analyses are taking a foothold in thoracic surgery, with sophisticated image processing being used more frequently to predict clinical outcomes [3,4,5,6]. In fact, some models utilizing radiomic features have outperformed traditional clinicopathologic models [7], demonstrating the potential of these techniques for improved prediction of postoperative survival.

Most models that predict long-term survival after esophagectomy rely on demographic, clinical, and pathologic variables without consideration of radiographically derived features [8,9]. However, CT scans contain an extensive amount of information related to bone, fat, and muscle, collectively known as body composition. Although body composition has been studied as a prognostic variable in other cancers [10,11], it remains relatively unexplored in esophageal cancer. There are other radiomic variables that have predictive capabilities after esophagectomy, such as sarcopenia and myosteatosis [12,13]. However, and quite importantly, most radiomic variables calculate tissue composition based on single images at a single anatomical location (i.e., from a single CT scan slice). To our knowledge, only one study has evaluated a survival model incorporating radiomic, PET, and body composition variables [14], finding sarcopenia to be an independent predictor of survival.

Quantifying various body composition tissues depicted on CT scans using traditional manual approaches can be technically challenging and time-consuming. Moreover, practical consideration often necessitates compromises in the calculations, such as using a single slice with cutoffs to calculate sarcopenia, which does not account for intermediate values of body tissues in other locations. To overcome some of these challenges, we developed computer software to automatically segment three-dimensional body composition from CT images [15], allowing for extensive and accurate quantification. Compared to other methods, our software computes a larger number of variables and provides precise values for each body tissue type [16]. In this way, a more comprehensive assessment of body composition can be made compared to evaluating sarcopenia from a single image slice.

In this study, we built a post-esophagectomy survival model that incorporates a comprehensive set of body composition features from pretreatment CT scans, tumor radiomic features from preoperative CT and PET-CT scans, and clinical features (including pathologic stage). The model was designed to: (1) predict post-esophagectomy survival, (2) assess the impact of body composition features on the model performance, and (3) compare the performance of our model to other models.

## 2. Patients and Methods

### 2.1. Study Population

This study was approved by the University of Pittsburgh Institutional Review Board (IRB #: STUDY20100305) on 5 February 2021. Our dataset was developed from a prospectively collected database of all patients undergoing esophagectomy at the University of Pittsburgh Medical Center (UPMC) between 2008 and 2021. Inclusion criteria were patients who underwent esophagectomy for esophageal cancer, had available preoperative PET-CT and CT scans, and did not have preoperative chemotherapy or radiation. Data from 301 patients were de-identified and re-identified with a unique study ID number by an honest broker, and after removing patients with missing information, 183 patients ultimately met the inclusion criteria. Demographic (age, race, gender), clinical (stage, height, weight, smoking history), survival, and radiologic data were collected. CT scans were used to generate radiomic, tumor, and body composition features (described below). PET scans were used to generate SUV features, Total Lesion Glucose (TLG), and Metabolic Tumor Volume (MTV). Patients were followed for a mean of 31 months (0.1–132 months) post-surgery, and subjects without follow-up survival data were censored.

### 2.2. Image Acquisition

CT scans were performed over 13 years using Discovery STE (GE Healthcare, Waukesha, WI, USA) or Siemens Biograph scanners. The acquisition parameters are as follows: 120 kV or 130 kV, 60 mAs to 444 mAs, reconstruction field of view 275 × 450 mm, and image thicknesses ranging from 2–5 mm.

### 2.3. Image Features

Body composition: A three-dimensional (3-D) convolutional neural network (CNN) [15] was used to automatically segment five different body tissues depicted on CT scans, including visceral adipose tissue (VAT), subcutaneous adipose tissue (SAT), intramuscular adipose tissue (IMAT), skeletal muscle (SM) and bone (Appendix A). Each body tissue was quantified with three measurements: mass, volume, and density (measured in average Hounsfield units (HU)). Mass was estimated from CT HU values, where x is the HU value and y is the density (g/cm^3^) of a CT voxel [17] (Equation (1)).
(1)y=0.0011×x+1.136

Tumor radiomic features: The esophageal tumor contours were manually annotated on CT and PET-CT images using our in-house software system [18]. Three groups of image features were computed from pretreatment CT and PET scans based on the manual outlines:(1)Basic tumor features: (a) volume (mL), (b) density (HU), (c) mean diameter (mm), (d) maximum length (mm), and density based on average HU value.(2)CT-based tumor radiomic features: High-dimensional radiomic features (n = 500) were automatically extracted from the segmented regions on the CT images, which included summary, first order, shape, and gray-level co-occurrence features [19].(3)Tumor PET features: PET features were quantified by mapping tumor ROIs from the CT images onto PET images and then extracting functional characteristics, including maximum SUV, minimum SUV, average SUV, SUV entropy, SUV P75, PET metabolic tumor volume (MTV)/mL, and PET total lesion glycolysis (TLG).

### 2.4. Clinical Features

Clinical and demographic information included post-surgery survival time, pathologic stage, height, weight, BMI, age, sex, race, and smoking status (current-smoker, former-smoker, or never-smoker).

### 2.5. Statistical Analysis

Cox proportional-hazard model was used to perform survival analysis. All variables were initially evaluated by univariate analysis. Then, multivariate Cox proportional-hazard models were created using backward-stepwise regression. The final model kept all variables with a *p*-value less than 0.05. The primary performance metric for the Cox proportional-hazard model was the concordance index (c-index). Independent time-point survival classification was performed on subsets of the cohort at three time points: one-year, three-years, and five-years post-esophagectomy. Patients were included into the time points if their survival time was greater than the time of consideration or their date of expiration was known (not censored). Models constructed include the logistic regression modeling with L1 penalty [20], the Naïve Bayes model with a Gaussian prior [21], the random-forest model with the Gini impurity index [22], and the support vector machine (SVM) with the radial basis kernel function [23,24]. Models were trained and tested on both normalized and un-normalized data; the model with superior performance was selected. Z-score normalization was used to standardize the data.

Two sets of models were created for each time point. One model was created using just pathologic T-, N-, and M-stage variables, herein termed the “reference model”. The second model included stage variables in addition to the clinical and radiomic features. A multivariable model was constructed using stepwise forward logistic regression on each variable group (i.e., radiomic, tumor, PET, body composition, and clinical features). Intermediate variable models were then combined to create the final nested model using forward stepwise logistic regression. Training and validating the models were performed using 10-fold cross-validation. The performance metric was the area under the receiver operating characteristic curve (AUC/ROC). The DeLong test was used to assess the difference in performance between our model and the reference model, as well as to assess the impact of variables on the performance of the final model [25]. All statistics were performed in R 3.4.1 or Python. A *p*-value less than 0.05 was considered statistically significant.

## 3. Results

### 3.1. Body Composition

The cohort had 183 patients who had an esophagectomy but did not receive preoperative chemotherapy or radiation. Demographic and clinical information are summarized in Table 1. Appendix A lists the body composition and tumor features. There was no significant difference between body composition distribution based on any demographic categories.

### 3.2. Cox Regression Analysis

Sixteen of the 500 radiomic features had statistical significance in the univariate Cox proportional-hazard models for overall survival. Clinical features that contributed to mortality were advanced pathological T- and N-stage, being a former smoker, being older, and having low BMI (Table 2). VAT density was the only body composition from the univariate Cox proportional-hazard model that was statistically significant (Table 3). The final post-esophagectomy five-year survival model generated a concordance index (c-index) of 0.754, using eight variables, including four clinical features (race, BMI, pathological N-stage, pneumonia) and four body-composition features (bone density, muscle density, IMAT volume, SAT volume) (Table 4).

### 3.3. One-Year Survival

One-year survival had 147 patients included in the analysis. The SVM model for predicting one-year post-esophagectomy survival exhibited significantly higher performance than the logistic regression, Naïve Bayes, and random forest models (Appendix A). Specifically, the SVM model had an AUC of 0.817 (95% CI: 0.0.738–0.897) using five clinicodemographic (race, BMI, pathological n-stage, effusion, pneumonia), one radiomic feature (DV), 1 PET feature (SUV p75) and three body composition variables (bone density, bone mass, VAT density) (Appendix A). In contrast, the reference model, which used only pathological T-, N-, and M, yielded an AUC of 0.584 (95% CI: 0.461–0.725). The performance of the SVM model was significantly better than the reference model (Figure 1A, *p* = 0.0005). Furthermore, the full SVM model’s performance was significantly superior to the SVM model without body composition variables (Figure 1B, *p* = 0.0286), which produced an AUC of 0.725 (95% CI: 0.612–0.838).

### 3.4. Three-Year Survival

Three-year survival had 113 patients included in the analysis. The random forest model for predicting three-year post-esophagectomy survival exhibited superior performance compared to the other three models (Appendix A) with an AUC of 0.693 (95% CI: 0.594–0.792). This model incorporated six radiomic features (Mean intensity, 10th percentile intensity, root-mean-squared average intensity, summed average intensity, and grey-level co-occurrence matrix autocorrelation), four clinical features (race, pathological T-stage, pathological M-stage, smoking history), one PET feature (minimum SUV uptake), and three body composition features (bone mass, IMAT mass, IMAT volume) (Appendix A). The reference model achieved an AUC of 0.598 (95% CI: 0.491–0.706), and the SVM model did not show a significant difference (Figure 2A, *p* = 0.127). The performance of the full random forest model was not statistically different from that of the random forest model without body composition variables, which achieved an AUC of 0.678 (95% CI: 0.578–0.777) (Figure 2B, *p* = 0.629).

### 3.5. Five-Year Survival

The five-year survival had 99 patients included in the analysis. The SVM model for predicting five-year post-esophagectomy survival outperformed other models (Appendix A) with an AUC of 0.861 (95% CI: 0.783–0.938). This model incorporated four radiomic features (mean, root-mean-squared mean, median, and 10th percentile intensity), three clinical features (age, BMI, pathological T-stage), one tumor feature (mean diameter), and two body composition features (VAT mass and IMAT volume) (Appendix A). Compared to the reference model with an AUC of 0.731 (95% CI: 0.617–0.845), the SVM model’s performance was significantly better (Figure 3A, *p* = 0.022). The full SVM model, including body composition variables, performed significantly better than the model without body composition variables (*p* = 0.042), which had an AUC of 0.801 (95% CI: 0.711–0.891) (Figure 3B).

## 4. Comment

Although studies have explored the potential of radiomics as a prognostic tool for cancer stage and survival [26,27], the impact of body composition on postoperative survival has received limited attention. We used a prospectively collected database of patients who underwent esophagectomy and specifically selected patients who did not receive neoadjuvant treatments. This approach allowed us to accurately determine the “true” pathologic stage, as that is a well-known predictor of survival.

Our model, which incorporated clinicopathologic, radiomic, and body composition variables, showed significantly higher performance in predicting one-year and five-year survival compared to a reference model that only used pathologic T, N, and M. Body composition variables were found to be important predictors. Although our three-year model had a higher AUC than the reference model, their difference was not significant according to the DeLong test (Figure 2A). This lack of significance could partly be due to the conservative nature of the DeLong test, which becomes increasingly conservative in “nested” models [28].

Sarcopenia on a single CT slice has been studied as a predictor of clinical outcomes, as it is thought to reflect a subject’s nutritional status, general health, and physical activity. However, our approach using body composition variables derived from a full-body CT scan may provide a more comprehensive assessment of overall health. Our univariate Cox proportional-hazard models indicated that the adipose tissue-related features (VAT, SAT, and IMAT) were all marginally significant, while SAT volume, IMAT volume, bone density, and muscle density volume became significant (*p* < 0.05) predictors in the multivariate Cox proportional-hazards model. This suggests that increased subcutaneous and intramuscular adipose tissue volumes and muscle density are associated with decreased survival after esophagectomy, while increased bone density is associated with improved survival outcomes. The role of IMAT in cancer patients is not well understood, but some theory suggests that increased IMAT may be linked to metabolic risk factors, such as insulin resistance, as well as negative effects on immune pathways and wound healing [29].

Previous studies have utilized logistic regression as the primary method for their survival prediction models post-esopahgectomy [5,8,9]. However, we took a different approach by evaluating sophisticated machine learning methods and conducted a comprehensive comparison of the four most common machine learning methods. We found that SVM had the best performance, even with a relatively small number of predictors (~10). Although logistic regression is a simpler method for prediction modeling and performs well with a smaller set of variables, SVM generates a high-dimensional feature landscape and improved separation of classes when incorporating a broader array of features.

The importance of including non-significant variables in the multivariate analysis should be noted, as some of the variables may become significant when included with other predictors in the model. Traditionally, only variables that show significance in univariate analysis are included in the multivariate analysis. However, the suppression principle in statistics recognizes that the addition of a third variable can provide more insight into the relationship between an independent and dependent variable [30]. Therefore, we have included some variables in the multivariate models that did not show significance in the univariate analysis to better understand the complex relationships between variables.

This study has several limitations that should be acknowledged. First, our analysis only include patients who did not receive neoadjuvant treatment, which allowed us to have accurate pathologic staging as a reference for survival prediction. However, our analysis does not take into account postoperative treatments, although postoperative complications were included into models. While our cohort only included preoperative CT-scans, future work that includes not only preoperative, but also postoperative CT-scans could greatly improve our understanding of how changes in body composition will affect overall survival outcomes. Additionally, we did not consider other potential prognostic factors, such as tumor grade or location, although most of the patients had distal esophageal adenocarcinoma. The use of various CT protocols over a 13-year period also introduced a diverse set of images used for model training, which could affect the generalizability of our findings. While we used an automated body composition segmentation tool, its accuracy is not perfect, but we find the results encouraging and believe that future studies should explore this further. Finally, external validation and larger cohorts are needed to establish the robustness of our work.

Esophageal cancer is a highly lethal form of cancer, and identifying better predictors of survival can aid in guiding patients toward appropriate therapy. Our study demonstrates that body composition is a significant contributing factor to survival models, and these variables can be automatically derived from preoperative images. The inclusion of body composition variables, such as intramuscular adipose tissue volume, in survival models can provide a more comprehensive assessment of prognosis. This may lead to improved personalized treatment strategies and ultimately better outcomes for patients with esophageal cancer.

## Figures and Tables

**Figure 1 jcm-12-02106-f001:**
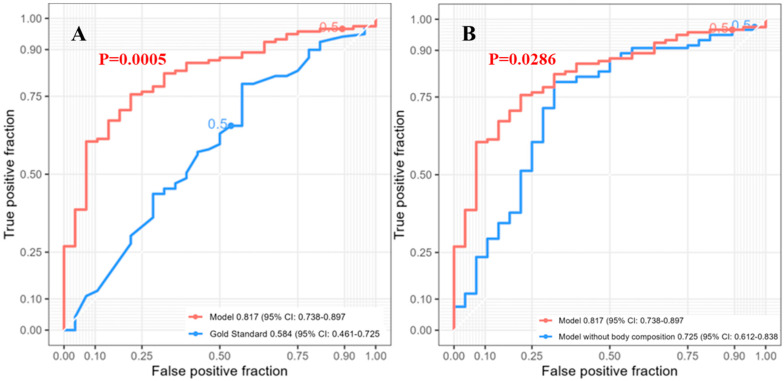
ROC curves of SVM models predicting one-year post-esophagectomy survival (n = 147). (**A**) Full SVM model and reference model. (**B**) SVM models with and without body composition variables. SVM = support vector machine. ROC = receiver operating characteristic.

**Figure 2 jcm-12-02106-f002:**
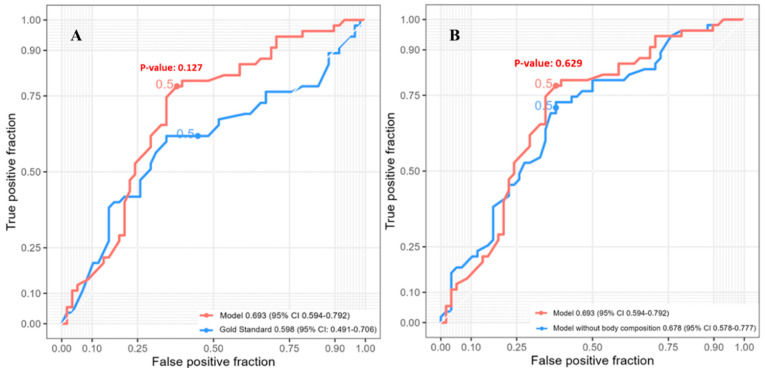
ROC curves of random forest models predicting three-year post-esophagectomy survival (n = 113). (**A**) Full random forest model and reference model. (**B**) Random forest models with and without body composition variables. ROC = receiver operating characteristic.

**Figure 3 jcm-12-02106-f003:**
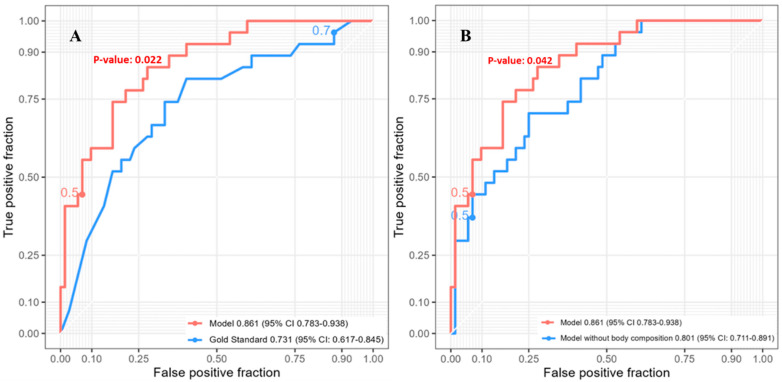
ROC curves of SVM models predicting five-year post-esophagectomy survival (n = 99). (**A**) Full SVM model and reference model. (**B**) Full SVM model and SVM model without body composition variables. SVM = support vector machine. ROC = receiver operating characteristic.

**Table 1 jcm-12-02106-t001:** Baseline demographics (n = 183).

Characteristic	Value
Age	6.7 ± 10.48
Height (cm)	171 ± 9.97
Weight (kg)	85 ± 21.20
Male	138 (75.41%)
Race	
White	179 (97.81%)
Black	3 (1.64%)
Asian	1 (0.05%)
Smoking	
Former-smoker	120 (65.57%)
Never-smoker	51 (27.87%)
Current-smoker	12 (6.56%)
Cancer Stage	
I	65 (35.52%)
II	31 (16.94%)
III	59 (32.79%)
IV	27 (14.75%)
Postoperative Complications	
Atrial Fibrillation	42 (22.95%)
Pneumonia	41 (22.40%)
Effusion	37 (20.22%)
Mean ± (SD); n (%)	

**Table 2 jcm-12-02106-t002:** Univariate Cox proportional-hazard models for clinical information.

Covariate	Coefficient	Hazard Ratio	Lower 95%	Upper 95%	*p*-Value
Age	0.024	1.024	0.004	0.044	0.021
BMI	−0.049	0.952	−0.089	−0.01	0.014
Height(cm)	0.009	1.009	−0.013	0.032	0.422
Weight(kg)	−0.009	0.991	−0.02	0.002	0.1
Sex					
Female	−0.081	0.922	−0.621	0.459	0.768
Male	0.081	1.084	−0.459	0.621	0.768
Race					
White	−0.484	0.617	−1.32	0.353	0.257
Black	1.476	4.373	−0.533	3.484	0.15
Asian	−21.885	0	−83.864	40.095	0.489
Smoking					
Never-smoker	0.03	1.031	−0.455	0.515	0.903
Current-smoker	−0.138	0.871	−0.614	0.337	0.569
Former-smoker	2.47	11.826	0.974	3.967	0.001
Pathological T-stage					
T1	−0.877	0.416	−1.398	−0.355	0.001
T2	0.028	1.029	−0.551	0.608	0.924
T3	0.579	1.784	0.118	1.04	0.014
T4	1.569	4.801	0.613	2.524	0.001
Pathological N-stage					
N0	−0.891	0.41	−1.376	−0.406	0
N1	0.391	1.479	−0.083	0.866	0.106
N2	0.796	2.217	0.119	1.474	0.021
N3	0.51	1.665	−0.112	1.132	0.108
Pathological M-stage					
M0	−0.714	0.49	−1.626	0.199	0.125
M1	0.714	2.041	−0.199	1.626	0.125
Postoperative Complications					
Atrial Fibrillation	0.175	1.191	−0.345	0.694	0.510
Pneumonia	0.447	1.563	−0.049	0.943	0.078
Effusion	0.432	1.540	−0.099	0.962	0.111

**Table 3 jcm-12-02106-t003:** Univariate Cox proportional-hazard models for image features.

Covariate	Coefficient	Hazard Ratio	Lower 95%	Upper 95%	*p*-Value
Tumor Features					
Tumor volume	0.003	1.003	−0.002	0.009	0.272
Tumor density	0.004	1.004	−0.002	0.010	0.192
Tumor mean diameter	0.025	1.026	−0.001	0.052	0.061
Tumor maximum length	0.003	1.003	0.000	0.006	0.093
PET Features					
Average SUV	−0.014	0.986	−0.087	0.059	0.710
Maximum SUV	−0.013	0.987	−0.068	0.041	0.633
Minimum SUV	0.030	1.031	−0.199	0.259	0.795
PET MTV/ml	0.000	1.000	0.000	0.001	0.135
PET TLG	0.000	1.000	0.000	0.000	0.270
SUV entropy	−0.129	0.879	−0.271	0.013	0.076
SUV P75	−0.012	0.988	−0.068	0.044	0.678
Body Composition Features					
VAT volume	−0.048	0.953	−0.120	0.024	0.191
VAT density	0.034	1.034	0.002	0.065	0.037
VAT mass	−0.047	0.954	−0.117	0.023	0.192
SAT volume	−0.021	0.979	−0.046	0.003	0.090
SAT density	0.013	1.013	−0.008	0.033	0.236
SAT mass	−0.021	0.979	−0.045	0.003	0.090
IMAT volume	0.023	1.023	−0.273	0.320	0.878
IMAT density	0.032	1.033	−0.006	0.071	0.096
IMAT mass	0.022	1.022	−0.261	0.305	0.878
SM volume	−0.035	0.966	−0.086	0.017	0.186
SM density	0.000	1.000	−0.026	0.026	0.984
SM muscle	−0.029	0.971	−0.073	0.014	0.183
Bone volume	−0.008	0.992	−0.259	0.244	0.951
Bone density	−0.004	0.996	−0.009	0.001	0.108
Bone mass	−0.024	0.976	−0.179	0.131	0.762

**Table 4 jcm-12-02106-t004:** Multivariate Cox proportional-hazard model for survival post-esophagectomy (n = 183).

Covariate	Coefficient	Hazard Ratio	Lower 95%	Upper 95%	*p*-Value
Race	0.363	1.438	0.124	0.603	0.003
BMI	−0.254	0.776	−0.356	−0.151	<0.0001
Staging					
Pathological N-Stage	0.533	1.705	0.315	0.752	<0.0001
Postoperative Complications					
Pneumonia	0.684	1.982	0.161	1.207	0.010
Body Composition					
Bone density	−0.006	0.994	−0.011	−0.001	0.031
Muscle density	0.062	1.064	0.015	0.109	0.009
IMAT volume	1.016	2.763	0.468	1.565	<0.0001
SAT volume	0.081	1.084	0.019	0.143	0.010
c-index: 0.754					

## Data Availability

Data is available by request to the corresponding authors (R.D. and J.P.).

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
