# Peer review of "CT-Derived Body Composition Is a Predictor of Survival after Esophagectomy"

_jcm, 2023, doi:10.3390/jcm12062106_

Round 1

Reviewer 1 Report

A novel approach of predicting the survival after esophagectomy. The analyses mentioned were based on preoperative CT or/and FDG PET-CT scans. Do the authors think that analyses based on post op CT examinations some months after surgery would affect the prediction models? Dividng the patients in different groups of post op complications can affect the prediction models? Maybe those issues should be mentioned in the discussion/comment session of the manuscript. Very interesting study!

Author Response

Response to Referee #1’s Comments

A novel approach of predicting the survival after esophagectomy. The analyses mentioned were based on preoperative CT or/and FDG PET-CT scans.

Do the authors think that analyses based on post op CT examinations some months after surgery would affect the prediction models?

Answer: While we have access to pre op CT, we do not have access to follow up post op CT imaging. However, we believe that including such post op examinations may improve models, as they would provide an updated estimate of body composition metrics post-surgery. We provided a discussion about this.

Dividing the patients in different groups of post op complications can affect the prediction models?

Answer: We thank the reviewer for this comment. We have performed additional experiments that have incorporated the most frequent post-operative complications in our cohort, namely, atrial fibrillation, pneumonia, and effusion. These variables were incorporated into the initial variable selection and were selected in the Cox proportional hazards model as well as the 1-year survival model.

Maybe those issues should be mentioned in the discussion/comment session of the manuscript. Very interesting study!

Reviewer 2 Report

This study examines the survival of 183 patients after esophagectomy. A computed model consisting of several variables (clinical, radiological, PET, tumor) is used to predict 1-, 3-, and 5-year survival, body composition plays an important role.

This is a very interesting study of a timely topic. I’m sure this innovative approach will gain more importance in the future. However, I have a few reservations and remarks:

Most of the limitations are mentioned in the limitations section, and I totally concur with those. Still, not incorporating the postop therapy is a huge downside, ways more important than the change in radiological and surgical techniques over the years. Is there a possibility of getting this data?

Could you explain why “only” 183 of 301 patients could be included? Was it missing data?

Did you record the reason for death? Cancer- and non-cancer-related?

How many patients are in the 1, 3 and 5-year survival-group?

P9, line235:  I understand your thinking; however, the response to neoadjuvant treatment is a predictor of survival

Author Response

Response to Referee #2’s Comments

This study examines the survival of 183 patients after esophagectomy. A computed model consisting of several variables (clinical, radiological, PET, tumor) is used to predict, 1-, 3-, and 5-year survival, body composition plays an important role.

This is a very interesting study of a timely topic. I’m sure this innovative approach will gain more importance in the future. However, I have a few reservations and remarks:

Most of the limitations are mentioned in the limitations section, and I totally concur with those. Still, not incorporating postop therapy is a huge downside, ways more important than the change in radiological and surgical techniques over the years. Is there any possibility of getting this data?

Answer: Although we do not have access to post-operative therapy for the patients included in the cohort, we do have access to post-operative complications. Per recommendation, we included additional analyses related to the most frequent post-op complications in our dataset, namely, atrial fibrillation, pneumonia, effusion, in the revision.

Could you explain why “only” 183 of 301 patients could be included? Was it missing data?

Answer: We updated the patient and methods section to include that patients excluded had incomplete information.  

Did you record the reason for death? Cancer- and non-cancer related?

Answer: Unfortunately, we do not have access to such data.

How many patients are in the 1, 3, and 5-year survival group?

Answer: We updated the patients and methods section to define the number of patients in each group. Additionally, the number of patients was included in the description of each ROC curve (figs. 1 -3).

P9, line 235: I understand your thinking; however, the response to neoadjuvant treatment is a predictor of survival

Answer: We agree with the reviewer upon this. As we explained, the subjects in our cohort did not have neoadjuvant treatment.

Round 2

Reviewer 2 Report

Thank you very much for the corrections. The main limitation re the postop treatment, remains. However, the novel method justifies a publication.